# Characterization of Synovial Fluid Components: Albumin-Chondroitin Sulfate Interactions Seen through Molecular Dynamics

**DOI:** 10.3390/ma15196935

**Published:** 2022-10-06

**Authors:** Natalia Kruszewska, Adam Mazurkiewicz, Grzegorz Szala, Małgorzata Słomion

**Affiliations:** 1Institute of Mathematics and Physics, Bydgoszcz University of Science and Technology, Kaliskiego 7 Street, 85-796 Bydgoszcz, Poland; 2Faculty of Mechanical Engineering, Bydgoszcz University of Science and Technology, Kaliskiego 7 Street, 85-796 Bydgoszcz, Poland; 3Faculty of Management, Bydgoszcz University of Science and Technology, Kaliskiego 7 Street, 85-796 Bydgoszcz, Poland

**Keywords:** molecular dynamics simulations, human serum albumin, chondroitin sulfate, synovial fluid, energy of binding, hydrophobic interactions, hydrogen bonds, ionic interactions

## Abstract

The friction coefficient of articular cartilage (AC) is very low. A method of producing tailor-made materials with even similar lubrication properties is still a challenge. The physicochemical reasons for such excellent lubrication properties of AC are still not fully explained; however, a crucial factor seems to be synergy between synovial fluid (SF) components. As a stepping stone to being able to produce innovative materials characterized by a very low friction coefficient, we studied the interactions between two important components of SF: human serum albumin (HSA) and chondroitin sulfate (CS). The molecular dynamics method, preceded by docking, is used in the study. Interactions of HSA with two types of CS (IV and VI), with the addition of three types of ions often found in physiological solutions: Ca${}^{2+}$, Na${}^{+}$, and Mg${}^{2+}$, are compared. It was found that there were differences in the energy of binding values and interaction maps between CS-4 and CS-6 complexes. HSA:CS-4 complexes were bound stronger than in the case of HSA:CS-6 because more interactions were formed across all types of interactions except one—the only difference was for ionic bridges, which were more often found in HSA:CS-6 complexes. RMSD and RMSF indicated that complexes HSA:CS-4 behave much more stably than HSA:CS-6. The type of ions added to the solution was also very important and changed the interaction map. However, the biggest difference was caused by the addition of Ca${}^{2+}$ ions which were prone to form ionic bridges.

## 1. Introduction

Many novel materials were designed based on ideas that mimic Nature. The approach became so common that a new interdisciplinary branch of science called biomimetics was developed. It is sometimes hard to produce innovative materials with properties nearing or matching the original naturally occurring biological systems. A great example of a natural biosystem characterized by hard-to-mimic properties is articular cartilage (AC). The friction coefficient of AC is very low—it is about ten times lower than ice on ice. Many experimental attempts to characterize the rheological properties of AC were performed. However, they have only been conducted in vitro, never taking into account all the components and features of the system. Physical measurements of the friction coefficient in synovial joints on standard industrial tribometers do not often give satisfactory results. The problems are the complicated shapes of the articular surfaces, load, variable speed, and direction of mutual movement of the surfaces or the variability of the roughness of cartilage [1]. In addition, during movement, the pressure inside the joint system also changes, resulting from changes in the volume of the joint’s capsule resulting from the work of the muscles and ligaments surrounding the joint. In addition, during movement, a temporary local load on the cartilage, and consequently, deformation of the articular surfaces appear. Because a layer of AC is very thin, the trabecular bone supporting the cartilage is also elastically deforming during load on bones under motion [2,3,4]. The mentioned factors make the physical measurements of the friction coefficient very difficult because many different aspects must be considered when planning the experiment. As a consequence, the results of measurements do not always correspond to the real values.

In [5], the authors presented an example of experimental measurement of the friction coefficient of a human shoulder’s joint during reciprocal loading in a pendulum testing device at a wide range of sliding speeds. The authors found that the friction coefficient remains very low (0.0015–0.006) for up to 24 hours of continuous loading. They claimed that the low friction coefficients observed in incongruent joints represent rolling rather than sliding friction. A possibility of lowering friction forces in the AC by altering the characteristic action of its components (phospholipid micelles trapped in the network of hyaluronic acid chains) from sliding to rolling was noted in [6]. Another example of friction measurements was reported in [7] for ACs collected from bovine knees. Using sliding pin-on-disc tribotester T-11 under physiological lubrication conditions, the authors measured friction coefficients versus wettability and obtained values in a range of 0.005–0.025.

It is not clearly defined which lubrication model best describes the lubrication mechanism of the AC. The phenomenon has been a subject of many theoretical considerations [8,9,10]. AC properties depend on the lubrication regime, which depends, in turn, on the amount of load on the system and the health of the joint [11]. Therefore, it is hard to propose an experiment that can imitate the in vivo system, but the puzzle can be solved by analyzing the interactions between system components. Because of all the complexities mentioned above, computer experiment methods appear to be very helpful in explaining many system behaviors [6,12,13,14,15].

A synovial fluid (SF) is present between the two opposite cartilages. It is composed mainly of hyaluronic acid (HA), lubricin, phospholipids, and various proteins. SF plays an essential role in synovial joint lubrication [16,17,18]. Changes in synovial fluid volume and composition reflect changes within the joints [19,20]. This is very important from the medical point of view because diseases can change the balance of the components of synovial fluid. For example, due to various diseases, the concentration of phospholipids and protein can be increased during the concentration, and the molecular weight of the HA can be decreased [21]. These medically important observations point out that the system should be studied as a whole; however, it is very hard due to its complexity. Recently, many research studies focused on subsystems, investigating interactions in pairs [12,13,22,23,24]. Following this approach, in the present paper, we study interactions between two SF elements: human serum albumin (HSA) and chondroitin sulfate (CS) immersed in a water environment. The importance of ions added to the solution is considered.

The first findings about the binding of CS to HSA were reported in [25]. The Authors performed an experiment using a spectropolarimeter with a UV circular dichroism attachment. Although they found that forces exist between the two molecules, the nature of the forces remained unknown.

In the present research, the intermolecular interaction in the system was studied to describe forces between HSA and CS. A computer simulation method using the molecular dynamics approach was used. A computer model of two complexes was created: HSA:CS-4 and HSA:CS-6. The molecular system was first prepared using the molecular docking (MDoc) method of dry molecules and next studied by molecular dynamics (MD) simulations in an aqueous environment (resembling physiological conditions). The number of intermolecular hydrogen bonds (HBo), hydrophobic–polar interactions (HP), ionic interactions, bridges (water and ionic), and energy of binding (EoB) between HSA and CS-4/CS-6 were calculated to determine the system’s dynamics. Moreover, the exact maps of contact were created to show the places of bindings.

## 2. Materials and Methods

### 2.1. Characterisation of Simulated Materials

CS stands as a pivotal component of synovial fluid. It is a highly negatively charged unbranched glycosaminoglycan (GAG) compound of N-acetylgalactosamine and glucuronic acid [26]. Two variants of CS are found in human joints: chondroitin sulfate IV and VI (denoted as CS-4 and CS-6, respectively). The difference between both CS types is presented in Figure 1.

The relative concentration of CS-4 versus CS-6 in joints is associated with cartilage calcification, while the calcification is related to histological degeneration of the joint [27]. In healthy cartilage, the concentration of both components is similar. In contrast, fully calcified cartilages usually have only CS-6 present in the synovial fluid [28]. An assessment of the influence of CS on the frictional–compressive properties of articular cartilage was presented in [29]. The authors deduced that the phenomenon of counter-ion condensation onto the CS chains influences the thermodynamic and frictional-compressive properties of the cartilage system.

This condensation tendency changes with the concentration of ions in the solution. An increase in condensation decreases the electrostatic friction between the chains and their resistance against gliding. Moreover, they demonstrated that the hydration shells of the counter-ions become smaller, which diminishes the resistance of the chains against compression. Their study concluded that at the physiological salt concentration chondroitin sulfate solutions possess optimal frictional–compressive properties. The CS molecules immersed in the SF are exposed to contact not only with ions but also with other molecules, such as proteins [30]. Therefore, the direct non-covalent intermolecular interactions appearing in the system, and indirect ones created by water and ionic bridges, could be of great importance for the rheological and tribological properties of the joint.

Experimental results indicate that CS is an effective lubricant for cartilage under mixed-mode lubrication conditions [31,32]. In [32], the authors speculated that the cartilage tissue might have a specific affinity for lubricants with negatively charged groups and hydroxyl groups on GAG chains, which may help them adsorb better to the cartilage surface, providing effective lubrication. Moreover, in [33], it was shown that HA or CS, when used alone, significantly lower the friction torque and dissipated energy of the fretting interface, which reduces the damage to the articular cartilage. It was also shown that HA and CS combined provided even better cartilage layer protection.

HSA is the most abundant protein in the SF. It is built on a single chain of 585 amino acids. It contains three homologous domains: I, II, and III [34]. Domains include residues (amino acids) as follows: Domain I: 5–197, domain II: 198–382, and domain III: 383–569. Each domain comprises two sub-domains termed A and B (IA: 5–107, IB: 108–197, IIA: 198–296, IIB: 297–382, IIIA: 383–494, and IIIB: 495–569), see Figure 2. The HSA regions responsible for the binding of ligands are known as Sudlow’s Site I and II and are located in subdomain IIA and IIIA, respectively. The heme binding site is located in subdomain IB [35,36,37].

HSA shows characteristic binding and transporting properties with fatty acids [38,39], steroids [40], bilirubin [41], ions, and many other molecules. It can also interact with HA. The strength of the interactions depends on, e.g., the amount of ions provided in the system, and affects the rheological and tribological properties of the cartilage [12]. HSA:HA-based complexes are good lubricants, and they considerably lower the friction coefficient [42,43]. New thin-film materials based on albumin and HA have been proposed to be used in biomedicine and cosmetics due to their adhesive properties [44]. Albumin-based nanomaterials are also proposed to be used in drug delivery and many other biomedical applications because, as natural agents, they have high biosafety and biodegradability [45]. Despite HSA and HA having a total negative charge under the physiological conditions, positively charged amino acids in albumin favor interactions with the ionized carboxylic groups in the HA [12]. On the other hand, CS is more negatively charged than HA due to the content of the sulfate group; thus, information about the lubrication properties of HSA:CS complexes could be valuable. Accordingly, to the author’s knowledge, no experiments study this complex’s friction properties. Moreover, the information about the structural features of HSA:CS molecular complexes and their intermolecular interaction characteristics [46] is still limited.

### 2.2. Molecular Dynamic Simulation Details

All-atom computer simulations of the model biosystem consisting of HSA and CS-4 or CS-6 molecules were performed. Thus, interactions inside two complexes (HSA:CS-4 and HSA:CS-6) were studied and compared to determine if a place of the sulfate group in the GAGs influences the binding properties.

In a first step, a molecular unbiased docking method was used to find the most energetically optimal places where CS-4 and CS-6 attach to the HSA. Next, 10 from the energetically best-docked structures (sorted from the strongest connection to the weakest connection), with added water solution of chosen ions (Na${}^{+}$, Mg${}^{2+}$, Ca${}^{2+}$ and Cl${}^{-}$), were subjected to MD simulations. Each realization had a system docked to a different part of HSA; as such, it represented different initial conditions. Both simulations (MDoc and MD) were performed using YASARA molecular modeling software [47].

Chemical structures of single units of CS-4 and CS-6 were obtained from Pubchem and modified to obtain 24 unit chains (around 8 kDa—longest chain allowable to dock with the modeling software used). The HSA structure was taken from the Protein Data Bank (PDB code: 1e78). Homological modeling using FASTA was performed before docking to fill in atoms missing in the PDB file.

To obtain the most stable complexes of CS ligand docked to HSA, the VINA method [48] was used with their default parameters and point-charge force field [49] initially assigned according to the AMBER14 force field [50]. Next, the system was damped to mimic the less polar Gasteiger charges used to optimize the AutoDock scoring function. A flexible receptor and ligand approach was chosen while docking. In both cases (HSA:CS-4 and HSA:CS-6), 10 of the strongest bound distinctive complexes which differ in the position of GAG docked to HSA with −10 kcal/mol free energy of binding were prepared for MD simulation.

Before MD simulations were conducted, optimization of the hydrogen bonding network was performed to increase the solute stability and pK${}_{a}$ prediction to fine-tune the protonation states of the protein residues at the given pH=7.4 [51,52]. Optimization was based on three steps as follows: first, pK${}_{a}$ prediction was carried out to consider the influence of the pH on the hydrogen bonding network; next, nonstandard amino acids and ligands were fully accounted for with the use of a chemical knowledge library in SMILES format; finally, the SCWRL algorithm was used to help find the globally optimal solution [53].

Both complexes were immersed in one of the three aqueous 2% salt solutions, NaCl, CaCl${}_{2}$, or MgCl${}_{2}$. After necessary minimization of the model system to remove clashes, the simulation was run for 140 ns using the AMBER14 force field [50] for the HSA, GLYCAM06 [54] for CS-4 and CS-6, and TIP3P for water. The cut-off distance for the van der Waals forces was set to 10 Å [55]. For computing long-range interactions (e.g., electrostatic interactions), the Particle Mesh Ewald algorithm was used [56]. Simulations were performed in a temperature of 310 K and under the pressure of 1 atm (NPT ensemble) [52]. A Berendsen barostat and thermostat were used to maintain constant temperature and pressure (relaxation time of 1 fs) [57]. Periodic boundary conditions were applied to a box of size equal to 120×110×110 Å${}^{3}$. The equations of motion were integrated with multiple time steps of 1.25 fs for bonded interactions and 2.5 fs for non-bonded interactions. The time step between stored states of the systems was equal to 100 ps. Thus, the time series for 140 ns of simulations obtained 1400 save points.

In order to characterize the binding between the macromolecules, the energy of binding and the number of intermolecular interactions (direct and via bridges) were computed. The energy of binding, obtained using YASARA’s algorithm, is the value (negative in principle) of the change of the energy of the system due to binding between the receptor (HSA) and the ligand (CS). The lower the binding energy, the stronger interaction between the components will be. The binding energy equation, which is the energy needed to disassemble a whole system into separate parts (which is equal to the energy of binding but with a reversed sign), has a form Equation (Equation 1)
(1)$$E_{bind}=E_{p}^{r}+E_{p}^{l}+E_{s}^{r}+E_{s}^{l}-E_{p}^{c}-E_{s}^{c},$$
where $E_{p}^{r}$ and $E_{p}^{l}$ are potential energies of separated compounds, i.e., receptor: HSA (r) and ligand: CS (l), $E_{s}^{r}$ and $E_{s}^{l}$ are their solvation energies, $E_{p}^{c}$ and $E_{s}^{c}$ are the potential and solvation energies of receptor–ligand complex. The energy of binding was computed using YASARA macro md_analyzebindenergy with the assumption that the cost of exposing one Å${}^{2}$ to the solvent is 0.65 kJ/mol. According to the YASARA Manual, the energy of binding may be shifted by an unknown constant that depends on the receptor (so on the value of the parameter mentioned above); thus, this quantity can only be used to compare various protein–ligand affinities rather than an absolute value.

## 3. Results and Discussion

In order to check whether the simulation times were long enough, the stability of the simulation based on the root-mean-square deviation (RMSD) of modeled molecules was computed. Exemplary RMSD results (for two best-bound complexes) are presented in Figure 3, and the rest are included in the Appendix A.

In virtually every case, the RMSD oscillated around a value of 2.5–3, indicating that the system reached stability in the given simulation time. Furthermore, in most cases, systems stabilized near 40 ns; thus, during intermolecular interactions analysis, the averaged results in the range from 40 ns to 140 ns were taken.

Moreover, the HSA and CS mobility was analyzed by calculating the time-averaged root mean square fluctuation (RMSF) values of HSA:CS-6 and HSA:CS-4 complexes. The RMSF as a function of a number of the atom is presented in the Appendix A. In order to compare the stability of HSA:CS-6 and HSA:CS-4 complexes, a sum of the RMSF values over all atoms belonging to the specific parts of the HSA and CS, and over all ten realizations for a specific complex in a specific solution, is presented in Appendix A. In the case of HSA, the parts are the protein domains. In the case of CS, the 24 mers were divided into eight groups (each group then has three mers) to simplify the presentation of the results. The behavior of the RMSF maximums seen in Appendix A is very similar to those presented for pure HSA computed for the protein amino acids [58]. Comparison of the results for HSA:CS-6 and HSA:CS-4 complexes in Appendix A has shown that sum of the RMSF is almost always greater for the case of CS-6 than for CS-4. The greater difference can be seen in the case of the NaCl solution. This suggests that both complexes immersed in NaCl solution are the most unstable because atoms fluctuated more than in all other cases. In general, the peripheral parts of the molecules fluctuated more than the middle ones, but it is mainly seen in the case of NaCl. The difference between the two charts in Figure S5 for HSA atoms was greater for HSA:CS-6 complexes, especially in the case of MgCl${}_{2}$. For CS, the difference in the fluctuation of CS-6 atoms was greater than CS-4, especially in the case of NaCl solution (cf., Figure S5b). The most negligible differences were noticed for CaCl${}_{2}$, which indicates more stable complexes.

In order to check the functionally important region and atoms fluctuations versus ligand contacts, the RMSF of the atoms belonging to the best-bound domains (IIIA for HSA:CS-6; Sudlow’s site II, and IIIB for HSA:CS-4) are presented in Figure 4.

In the figure, four of the strongest bound places are marked by arrows. The colors of the arrows show which atom from HSA has bound to the atom coming from CS. In the HSA:CS-6 complex, Pro (carbon atom) made HP interactions with the carbon atom of CS (blue arrow). A single carbon atom from CS (red arrow) created a few HP interactions with HSA bounding two different regions of HSA with CS. Carbon atoms from Glu and Asp created Ca${}^{2+}$ bridges with carbon atoms of CS (yellow and orange arrows). The existence of HP and HBo interactions of CS with Lys475 (marked with red letters) which is indicated as the binding site of long-chain fatty acids can be very important [37]. In HSA:CS-4, Lys had a few interactions with the carbon atom of CS (blue arrow). Glu ionic bridged with carbon atoms of CS (orange and yellow arrows; C-Ca-C bridges). Aps ionic bridged with the oxygen atom of CS (green arrow, C-Ca-O bridge). Usually, binding places from CS have lower values of RMSF, but in the case of HSA, it was not a rule. Thus, a stabilizing effect of CS on HSA cannot be reported. Note that in Figure 4, HP interactions, HBo, and ionic bridges are shown, but water bridges have been omitted due to their very short duration. At almost every MD step, other atoms took part in creating these bridges. In Appendix A, the RMSF for the IIIA domain for the HSA:CS-4 complex is shown to compare the same fragments between the two best-bound complexes. It can be seen that different parts of HSA interacted with CS-4 in a different place than with CS-6, and mainly the interaction was with Lys and Glu. In general, RMSF values were lower for CS-4 for both HSA and CS molecules.

In the present paper, the focus of the study was on specifying the bonding place of CS to albumin. The changes in albumin conformations were not the subject of the present study as it would need longer simulation times. However, a preliminary analysis of the secondary structure of HSA was performed. Their oscillations as a function of time are presented in Appendix A. Moreover, the comparison of the secondary structures at the end of the simulation (at 140 ns) for the two CS isomers in different ionic solutions is shown in Appendix A. Only a slight difference can be seen in the percentage content of helices and turns. HSA bound to CS-6 has more turns and fewer helices than in the case of the HSA:CS-4 complex.

Electrostatic interactions are essential for the binding mechanism of HSA and GAG complexes [59]. Electrostatic potential maps of albumin (with and without the addition of ions) are presented in [12] (cf., Figure 3 therein). The authors have shown that the presence of Na${}^{+}$, Mg${}^{2+}$, and Ca${}^{2+}$ cations caused a much higher positive charge density that could be observed in the middle of the electrostatic potential map of the HSA. This way, a specific cavity was formed to which GAGs’ negatively charged groups have a better chance to bind. This cavity is larger for divalent ions Mg${}^{2+}$ and Ca${}^{2+}$ than for monovalent Na${}^{+}$. Some HSA domains are more likely to bind to GAGs than others; however, the binding map can be altered under the condition of a disease [59,60]. The binding mechanism is mainly due to ionic bonding, hydrogen bonding, and hydrophobic interactions [59].

### 3.1. Energy of Binding

The binding energies obtained from YASARA VINE for complexes after the MDoc procedure are presented in Appendix A. Based on the table, a list of domains of HSA bound to CS-4 and CS-6 is presented in Table 1. The list of complexes is ranked according to the increasing magnitude of the energies of binding after MD. Its values were averaged over time from 40 ns to 140 ns (40 ns is when stabilization of the complexes was assumed based on the energy and RMSD of the whole system, cf. Figure 3). While each of the ten best-docked complexes had undergone three separate simulations in different solutions (with the addition of CaCl${}_{2}$, MgCl${}_{2}$, and NaCl) to obtain one value of the energy of binding for comparison and sorting purposes, the three values of energy of binding were averaged. The docking ranks of the complexes showing the binding strength order before MD simulations are presented in parentheses in Table 1.

From the table, it can be concluded that in the case of the HSA:CS-6 complex, the second best-docked structure (#2) turns out to be the best after MD simulations in the solution. The CS-6 docked to a wide range of HSA domains, IA-IB-IIA-IIIA-IIIB, and the greater amount of interactions were with the IIIA domain. In the case of the HSA:CS-4 complex, a third docked structure (#3) was bound strongest after MD, and in this case, the CS-4 built contacts with IB, IIIA, and IIIB (the last one was the strongest). The IA, IB, IIIA, and IIIB subdomains formed the characteristic binding center described in [12]. The albumin domains, IB, IIIA, and IIIB, were reported as very important for the albumin transport function responsible for the heme binding site (IB), Sudlow’s site II (IIIA), and the thyroxine-binding site (IIIB) [35]. In addition, all three domains were present in the two first strongest bound complexes for both HSA:CS-4 and HSA:CS-6. A similar feature was reported for HSA:HA complexes [12]. Comparing average MDoc binding energy for all HSA:CS-6 and HSA:CS-4 complexes, the HSA:CS-6 was bound about 23% stronger than HSA:CS-4. However, comparing HSA:CS-6 to HSA:HA, the binding in HSA:CS-6 was about 11% weaker than for HSA:HA [12]. It is consistent with expectations because CS is more negatively charged than HA (CS has two negative groups: COO- and sulfate groups, but HA has only COO-); thus, also binding it with negatively charged albumin is weaker.

Snapshots of the HSA:CS-6 and HSA:CS-4 complexes in CaCl${}_{2}$ solution before and after 140 ns of MD simulation for best-bound complexes (#2 for HSA:CS-6 and #3 for HSA:CS-4) are presented in Figure 5. Similar Figures for MgCl${}_{2}$ and NaCl solutions are presented in Appendix A.

Generally, best-bound complexes after MDoc are not necessarily best-bound after MD simulations. This statement can be explained by the influence of water solution, which changes both docked molecules’ electrostatic map (and conformation). Adding ions into the solution can provide charge inversion and ion-bridge formation [61,62].

In Figure 6, the energy of binding for different complexes is shown. The values are averaged over time from 40 to 140 ns of MD simulation with a doubled standard deviation that reflects fluctuations of the energy values.

After MD simulations, the energy of binding for HSA:CS-4 is definitely of lower value than for HSA:CS-6; thus, the binding is more stable for the HSA:CS-4 complexes. Computation of averages for all the energy values over HSA:CS-6 and HSA:CS-4 provides the information that, after MD, complex HSA:CS-4 is about 15% stronger bound than HSA:CS-6, thus the situation is opposite than before MD simulations (cf. Appendix A). In about half of the cases (six for CS-6 and four for CS-4), adding CaCl${}_{2}$ into the solution caused the most stable complexes. For 3 out of 10 CS-6 complexes, the highest affinity of CS to HSA was observed in the presence of Mg${}^{2+}$ ions and only 1 in the presence of Na${}^{+}$ ions. In the CS-4 isomer, the energy of binding was the lowest in three complexes for Mg${}^{2+}$ ions and in three complexes for Na${}^{+}$ ions. However, complexes with the addition of NaCl usually created weaker bound systems than CaCl${}_{2}$ and MgCl${}_{2}$. It was also confirmed by RMSF analysis (cf. Figure 4).

The energy of binding as a function of time for best-bound complexes is shown in Figure 7.

The charts were plotted for best-bound cases (#2 for HSA:CS-6 and #3 for HSA:CS-4). The vertical lines show values averaged from 40 ns to 140 ns (thus, after stabilization of structures in MD). In both cases, structures with the addition of Ca${}^{2+}$ to the solution are bound stronger than in the rest of the cases. It is very interesting that for HSA:CS-6, all averaged energies were similar. The trend is only seen for HSA:CS-4 and accords with the expectation which considers the cavity’s role in the electrostatic potential map of the HSA (described earlier in Section 3) and cations bridges formation in the macromolecules binding. This confirms the prominent role of Ca${}^{2+}$ in the binding.

The binding of the complexes occurs through intermolecular interactions. The interactions can be direct, when some forces appear between two atoms at a specific, close distance, or indirect, when another atom mediates the binding (creating bridges).

### 3.2. Intermolecular Interactions

The numbers of HP interactions (between hydrophobic atoms) and HBo were calculated with the algorithm described previously [12,63]. According to the YASARA definition, the HBo is formed when the hydrogen bond energy is greater than 25% of the optimum value for interaction 25 kJ/mol and equals 6.25 kJ/mol. The exact formula is described in the YASARA Manual [51] and previously in [12,63].

The numbers of intermolecular interactions between HSA and CS-6 or CS-4, also averaged over 40–140 ns for complexes sorted by averaged energies of binding after MD are shown in Figure 8. The numbers of the complexes correspond to the ones presented in the first column of Table 1 (before parenthesis).

The main observation from Figure 8 is that the charts are not much different. It can be seen that the number of HBo is slightly greater for complexes characterized by lower energy of binding, thus a stronger bound. The same can be seen for HP interactions in HSA:CS-4 but not in HSA:CS-6. In general, HSA:CS-4 has created more HBo and HP interactions. It also has a more varied plot of ionic interactions than HSA:CS-6. The number of ionic interactions is the lowest among the others. However, thanks to their electrostatic origin (the electrostatic force of attraction governs them), they are the strongest, so they are also important. HP interactions are also usually stronger than HBo. Thus, it is hard to determine which of the three non-covalent interactions influences most of the HSA:CS binding. An additional important observation is that the number of ionic interactions in most complexes is greater for the NaCl solution (cf., Figure 8e,f). Analyzing the influence of the ions added to the solution, the most visible is the prevalence of the number of HBo in the case of best-bound complexes in NaCl solution. This can be caused by Na${}^{+}$ having the lowest ionic strength among the three ionic solutions (NaCl, CaCl${}_{2}$, and MgCl${}_{2}$). As a result, it does not have such strong adsorption properties on the surface. As a result, the HBo are formed more often in NaCl than in ionic interactions or ion bridges preferentially formed in divalent ion solutions.

All of the above observations confirm that the binding affinity cannot be related to only one type of interaction but is a result of many different types of interactions.

### 3.3. Water and Ionic Bridges

The number of bridges created by water molecules and ions between HSA and CS for different complexes HSA:CS-6 and HSA:CS-4 are shown in Figure 9. The water (or ion) bridge is created when one water (or ion) molecule forms an HBo (or ionic interaction) to HSA and another one to CS.

By analyzing both Figures, it can be seen that HSA:CS-4 complexes characterize a greater number of water bridges, while the HSA:CS-6 complexes have more ionic bridges. Water bridges are very important for energy in binding. Their number usually decreases with the rank of the complex. It can suggest that direct HBo (cf., Figure 8) and indirect ones (i.e., mediated by water bridges) are the most important for HSA:CS binding. HSA:CS-4 has more intermolecular HBo, HP, and ionic interactions than ionic bridges. In the HSA:CS-6 case, in contrast, there are much fewer HBo, HP, and ionic interactions than ionic bridges, even though the greater number of ionic bridges could not make up for energy shortages caused by the lower number of HBo, HP, and ionic interactions. The ionic bridges built by Ca${}^{2+}$ are usually created between sulfur and carbon atoms in the case of the HSA:CS-6 complex and between carbon and carbon in the case of the HSA:CS-4 complex.

An important observation should be made for CS-6 complex #1, in which every direct intermolecular interaction type (HBo, HP, ionic) has a greater number of interactions in the case of Na${}^{+}$ than Ca${}^{2+}$ and Mg${}^{2+}$. It also has slightly more water bridges, but at the same time, it has a much smaller number of ionic bridges. Taking into account that the energy of binding for Na${}^{+}$ is greater than for Ca${}^{2+}$ and Mg${}^{2+}$ and that the complex is weaker bound for Na than for Ca and Mg, it can be stated that the ionic bridges are of great importance for the stability of HSA:CS-6 complexes. The visible difference in the effect of Ca${}^{2+}$ and Mg${}^{2+}$ for ionic bridge formation is suggested to be due to the lower hydration of Ca${}^{2+}$ [12]. The influence of Ca${}^{2+}$ and Mg${}^{2+}$ on albumin binding can be explained by the fact that albumin interacts with it. HSA has almost no interaction with Na${}^{+}$ ions [64].

While hydration properties are crucial for lubrication, a number of HBo created between HSA:CS complexes and the water molecules was analyzed (see Appendix A). In nearly every case, HSA:CS-6 complexes created more HBo with water thanHSA:CS-4 did. It confirms better binding between HSA and CS-4 (less space for water molecules to interact) than CS-6. A greater difference can be seen for complexes immersed in CaCl${}_{2}$ solution when water created a lesser number of HBo with both complex types. This, together with observations of a much higher number of ionic bridges and smaller energy of binding in the case of Ca${}^{2+}$, is evidence of stronger binding.

### 3.4. Maps of Interactions

In Figure 10, the maps of HBo distribution between different groups or atoms are presented. The results were summarized for all HBo interactions found between 40 and 140 ns of simulations.

In each case, most HBo interactions were created between SO${}_{4}^{-}$ and Arg or SO${}_{4}^{-}$ and Lys, thus positive-charged amino acids. Moreover, the COO${}^{-}$ group has bound to Lys. From CS, most bindings had the group of SO${}_{4}^{-}$ and for HSA—Lys and Glu. In addition to these expected results, a distinctive impact was noticed for: Gln, Asp, Tyr, and Ser. Moreover, O, O3, and O14 interactions with Glu are also clearly visible in Figure 10. Interestingly, COO${}^{-}$ interacted with Arg using HBo in the case of NaCl water solution; also, O13-O14 with Glu is better visible when NaCl is present (cf., Figure 10c,f). There are no clearly visible differences between CS-6 and CS-4, except in the case of CS-6 + Na${}^{+}$ where interactions between SO${}_{4}^{-}$ and Glu and His are noticeable.

In Figure 11, similar maps were created, but for the number of water bridges between HSA and CS.

The water bridges’ maps look different from those created for direct HBo interaction. The leading roles of SO${}_{4}^{-}$ and Lys did not change, but in this case, interactions created by Glu are much more visible. The role of Glu in building water bridges is similar to Lys. SO${}_{4}^{-}$ made interactions via water bridges with Lys, Glu, Asp, and Arg more frequent. A slight difference can be seen in the case of the NaCl solution, where Glu+O14 is marked more strongly than in the rest of the cases. Moreover, SO${}_{4}^{-}$ + Arg and Asp interactions are more frequent for NaCl than for the rest. On the other hand, differences between CS-4 and CS-6 are not visible.

## 4. Conclusions

In the present paper, interactions between HSA and CS-4 and between HSA and CS-6 were studied. In both cases, HSA can form stable complexes, but RMSD and RMSF indicated HSA:CS-4 as behaving much more stably. The binding strength and interaction distribution also differed for HSA:CS-4 and HSA:CS-6 complexes. It can be explained by different intramolecular interactions in the two isomeric forms of CS, which also causes their different conformation in the water solution [65]. MD simulations have shown that CS-4 has a greater affinity for binding to HSA than CS-6 does. Because the percentage content of the two types of CS differs for healthy and ill cartilage, it can be deduced that the lubrication properties of SF containing CS-6 and CS-4 will be different from the ill ones which contain only CS-6. CS-6 is characterized by worse stability when interacting with HSA. Additionally, it can be inferred [32] that the cartilage tissue may have a specific affinity for lubricants with negatively charged groups and hydroxyl groups such as in CS. This may help them adsorb better to the cartilage surface, providing effective lubrication. Thus, the lack of stronger binding provided by the CS-4 type of CS isomer in ill cartilages can explain worse lubrication properties.

The ions contained in the solution are also essential and change the interaction map of the HSA:CS complexes. It is especially seen in the case of complex #1 of HSA:CS-6. Despite having more HP and HBo interactions where a solution of NaCl was concerned, the ionic (Ca) bridges balanced the energies of binding, indicating that HSA:CS-6 with NaCl was weaker bound than with CaCl${}_{2}$. Thus, the availability of Ca${}^{2+}$ for ionic interaction formation via bridges seems to be the most important. The presence of Ca${}^{2+}$ (and also Mg${}^{2+}$, but less so) amplifies the binding mechanism in the case of HSA:CS, which was mainly associated with the presence of locally positively charged sites (mainly Lys and Glu). The three domains, very important for the albumin transport function, i.e., IB (heme binding site), IIIA (Sudlow’s site II), and IIIB (thyroxine-binding site), were presented in the binding in most of the complexes, especially in those characterized by the strongest binding [35].

Analyzing the obtained results, the similarity of HSA:CS binding to HSA:HA binding is evident. The interaction strength was slightly smaller for HSA:CS, but the influence of ions on the binding was similar [12]. Because the addition of HSA and CS separately decreases the friction coefficient [29,42], a complex of HSA:CS could give better results for lubrication properties, similar in the case of HSA:HA [12,42].

To the authors’ knowledge, while writing the paper, the findings have not yet been confirmed by experimental data. However, the authors hope that the presented study can inspire other research groups to undertake such an endeavor.

Because understanding the nature of interactions between HSA and CS can be a stepping stone to explaining the lubrication properties of AC, the information contained in this work can be potentially applicable to designing new biomaterials characterized by specific rheological properties.

## Figures and Tables

**Figure 1 materials-15-06935-f001:**
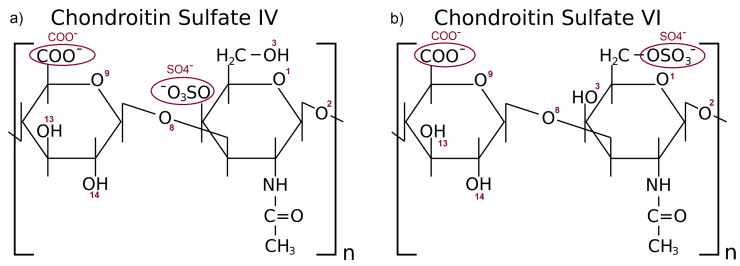
Schematic representation of the CS-4 (**a**) and CS-6 (**b**). The numbers of oxygen atoms and functional groups involved in forming hydrogen bonds are labeled with red letters.

**Figure 2 materials-15-06935-f002:**
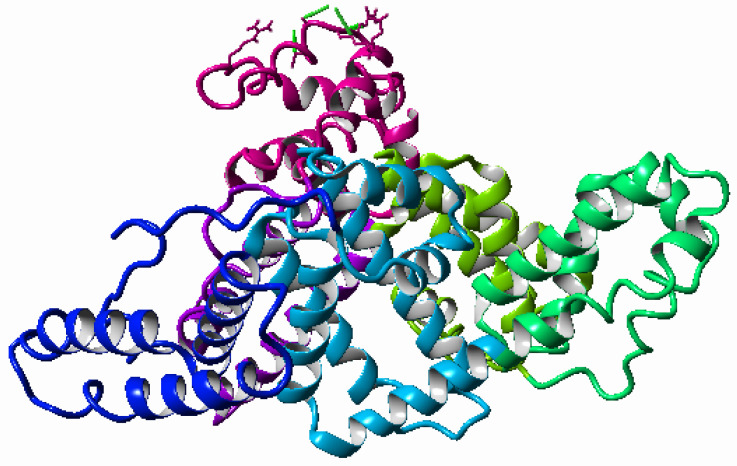
Ribbon representation of albumin in YASARA. Albumin domains are colored as follows: IA-pink, IB-violet, IIA-light green, IIB-green, IIIA-light blue, and IIIB-blue.

**Figure 3 materials-15-06935-f003:**
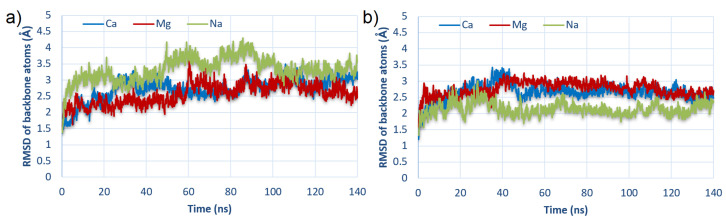
RMSD of backbone atoms of HSA:CS complexes as a function of time for the strongest bound complexes: (**a**) #2 for HSA:CS-6; and (**b**) #3 for HSA:CS-4 (cf., Table 1).

**Figure 4 materials-15-06935-f004:**
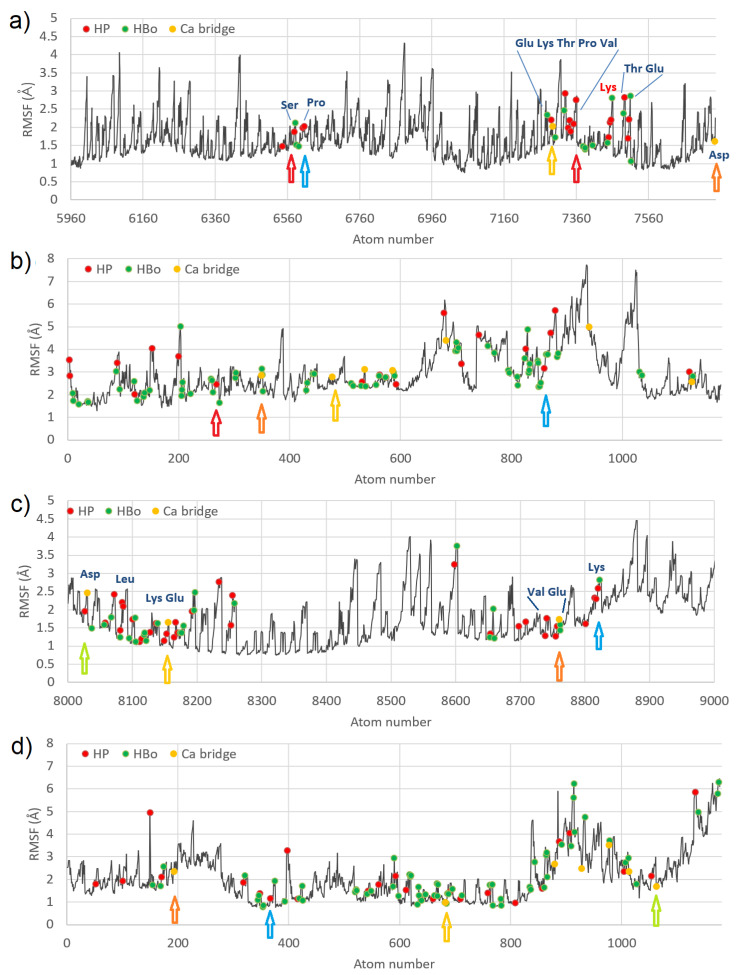
RMSF for best-bound domains of HSA (**a**,**c**) and CS (**b**,**d**) for complexes #2 HSA:CS-6 (**a**,**b**), and #3 HSA:CS-4 (**c**,**d**); the places of specific interactions are marked with dots. In the case of HSA:CS-6, the best-bound domain was IIIA (**a**), and for HSA:CS-4, the best-bound domain was IIIB (**c**).

**Figure 5 materials-15-06935-f005:**
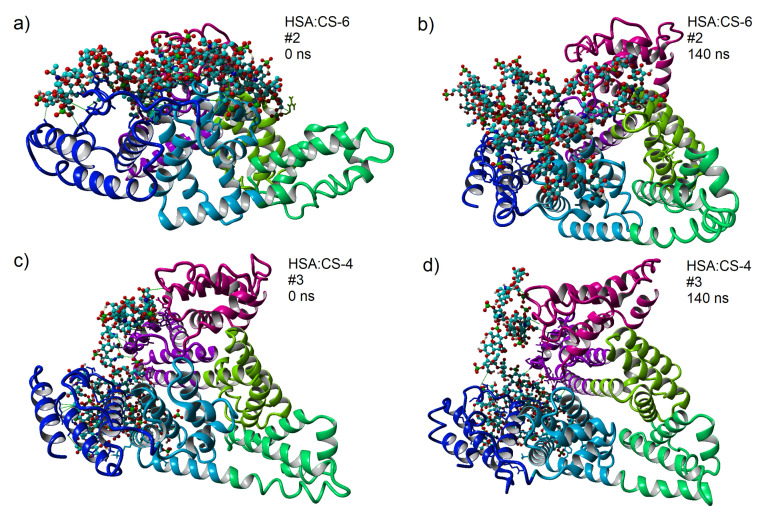
3D structures of HSA:CS-6 (**a**,**b**) and HSA:CS-4 (**c**,**d**) complexes for the strongest bound complexes after MD in CaCl${}_{2}$ solution (solution is transparent on the picture). HSA domains are colored as follows: IA-pink, IB-violet, IIA-light green, IIB-green, IIIA-light blue, and IIIB-blue. In CS-4 and CS-6, light blue atoms represent carbon; dark, blue nitrogen; red, oxygen; green, sulfur; and white, hydrogen. Snapshots are taken using YASARA software (**a**,**c**) before and (**b**,**d**) after 140 ns of MD simulations [47]. After a closer look at these pictures, green and pink lines can be observed, which show HP and ionic intermolecular interactions, respectively, and also yellow lines, which show intramolecular HBo inside CS.

**Figure 6 materials-15-06935-f006:**
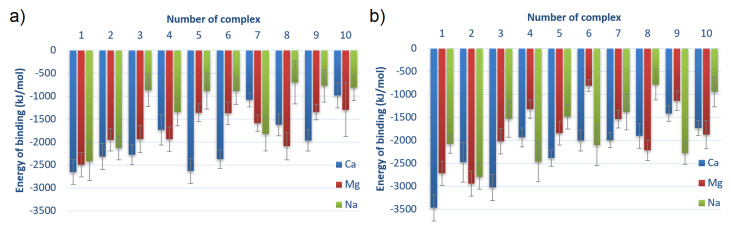
Averaged energy of binding for different complexes for (**a**) HSA:CS-6 and (**b**) HSA:CS-4. Complexes are sorted from lowest to highest averaged energy of binding after MD simulations; thus, the strongest bound are first (cf. Table 1). Error bars present doubled STD.

**Figure 7 materials-15-06935-f007:**
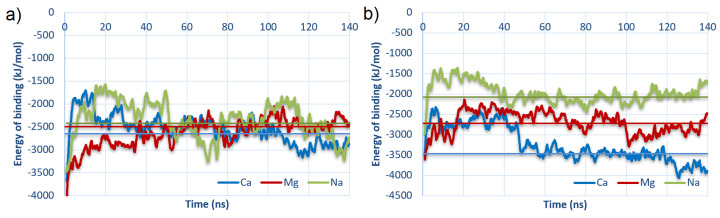
Energy of binding as a function of time for the strongest bound complexes: (**a**) #2 for HSA:CS-6; and (**b**) #3 for HSA:CS-4 (cf., Table 1). The vertical lines represent the average over the last 100 ns of MD.

**Figure 8 materials-15-06935-f008:**
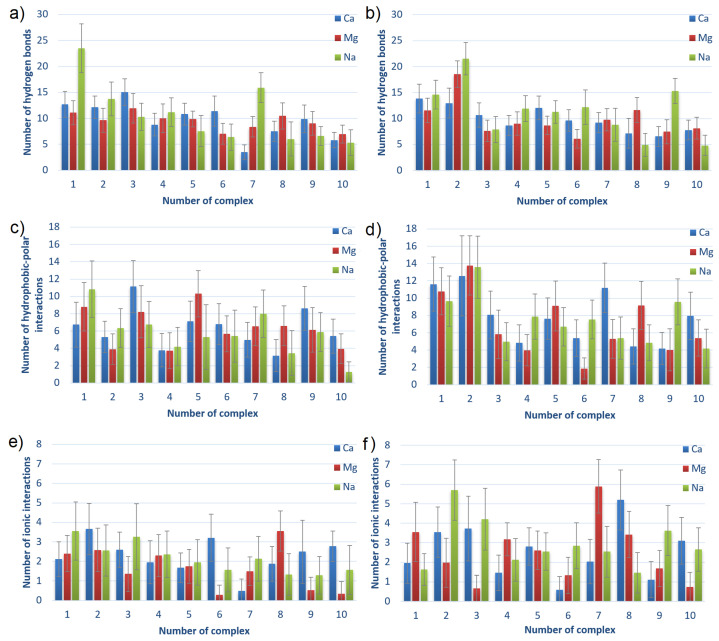
Intermolecular HBo, HP and ionic interactions for HSA:CS-6 (**a**,**c**,**e**) and HSA:CS-4 (**b**,**d**,**f**) complexes. The complexes are sorted from lowest to highest averaged energy of binding after MD simulations; thus, the strongest bound are first (cf., Table 1). Error bars present doubled STD.

**Figure 9 materials-15-06935-f009:**
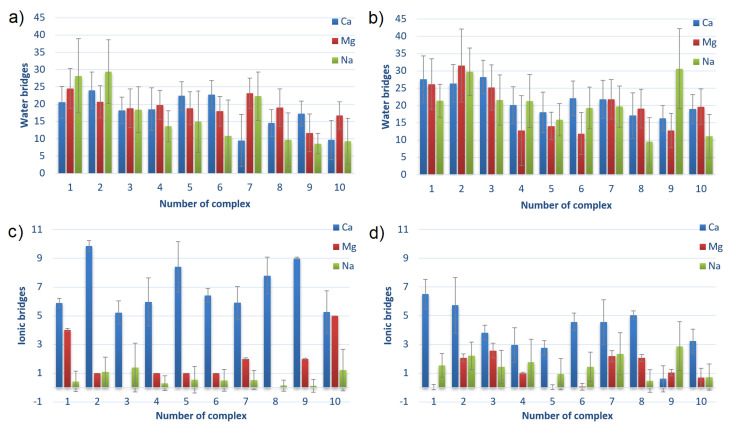
The number of hydrogen bonds mediated by water molecules (water bridges) and the number of ionic interactions mediated by cations (ionic bridges) between HSA and CS-6 (**a**,**c**) and between HSA and CS-4 (**b**,**d**). The complexes are sorted from lowest to highest averaged energy of binding after MD simulations; thus, the first is the strongest bound (cf., Table 1). Error bars present doubled STD.

**Figure 10 materials-15-06935-f010:**
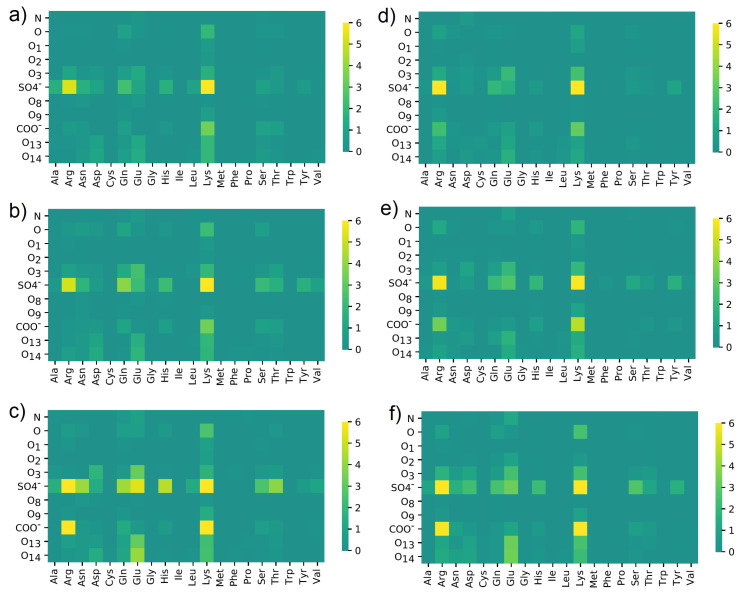
HBo distribution between different oxygen classes in (**a**–**c**) CS-6 and (**d**–**f**) CS-4; and different amino acids in HSA. Data were obtained in solutions containing: (**a**,**d**) NaCl; (**b**,**e**) CaCl${}_{2}$; (**c**,**f**) MgCl${}_{2}$. The maps present a number of interactions. The denotation of the atoms and functional groups is presented in Figure 1.

**Figure 11 materials-15-06935-f011:**
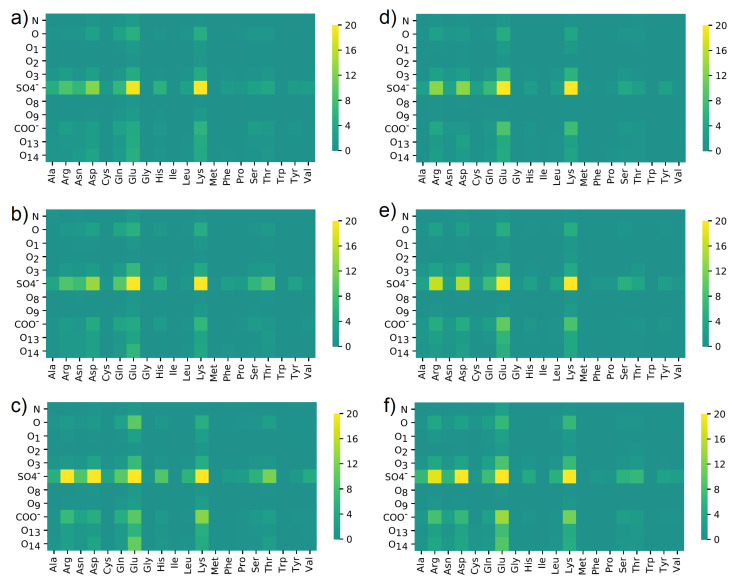
Water bridges distribution between different oxygen classes in (**a**–**c**) CS-6 and ((**d**–**f**) CS-4 and different amino acids in HSA. Data were obtained in solutions containing: (**a**,**d**) NaCl; (**b**,**e**) CaCl${}_{2}$; (**c**,**f**) MgCl${}_{2}$. The maps present a number of interactions. The denotation of the atoms and functional groups is presented in Figure 1.

**Table 1 materials-15-06935-t001:** Binding ranks of HSA-CS6 (up) and HSA-CS4 (down) complexes. The first column contains two numbers: rank after MD simulations averaged over EoB, and in parenthesis, rank after docking (and before MD simulations). The second column provides values of EoB with STD, averaged over the part of the trajectory after equilibration (and for all ions). The strongest connected domains are marked with bold letters.

HSA-CS6 Rank	EoB (kJ/mol)	HSA Binding Sites
1(2)	−2522 ± 339	IA-IB-IIA-**IIIA**-IIIB
2(6)	−2133± 301	IB-IIIA-**IIIB**
3(9)	−1694± 665	IB-IIIA-**IIIB**
4(7)	−1670± 388	**IA**-IIA-IIB-IIIA
5(1)	−1628± 792	IA-IIA-IIIA-**IIIB**
6(5)	−1542± 665	IA-IIA-IIIA-**IIIB**
7(4)	−1498± 399	IA-IIA-IIIA-**IIIB**
8(3)	−1472± 675	IA-**IIA**
9(10)	−1363± 550	IIA-**IIB**
10(8)	−1033± 453	**IA**-IB
**HSA-CS4 Rank**	**EoB (kJ/mol)**	**HSA Binding Sites**
1(3)	−2755 ± 624	IB-IIIA-**IIIB**
2(9)	−2737 ± 386	IB-IIA-**IIIA**-IIIB
3(8)	−2194 ± 702	IA-IB-**IIA**
4(10)	−1906 ± 556	**IA**-IB-IIA-IIIA
5(4)	−1904 ± 441	IA-IB-**IIA**-IIB
6(5)	−1641 ± 659	IB-IIA-IIB-**IIIA**-IIIB
7(7)	−1639 ± 374	**IA**-IB-IIA-IIIA-IIIB
8(2)	−1637 ± 674	IA-**IB**-IIIA-IIIB
9(1)	−1613 ± 531	IB-IIA-IIIA-**IIIB**
10(6)	−1516 ± 493	IB-IIIA-**IIIB**

## Data Availability

All data is available in the manuscript or upon request to the corresponding author.

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
