# Peer review of "Characterization of Synovial Fluid Components: Albumin-Chondroitin Sulfate Interactions Seen through Molecular Dynamics"

_materials, 2022, doi:10.3390/ma15196935_

Round 1
Reviewer 1 Report
This work studied the interactions between two different components of the synovial fluids, which is important for understanding the lubrication properties. Some questions need to be considered as follows:
1. The introduction part should be simplified.
2. Did the authors consider the hydration properties of the studied system, which is critical for lubrication?
3. The simulation results should be compared with some published experimental results.
Reviewer 2 Report
Kruszewska and co-authors studied the interactions between two types of chondroitin sulfate with human serum albumin by means of molecular docking followed by classical molecular dynamics simulations. The interaction between CS and HSA is important from two points of view, it can explain mechanism behind cartilage calcification and help prevent it, and serve as the basis in the development of novel materials with low friction coefficients. Interaction types (hydrogen bonds, hydrophobic interactions, water and ionic bridges) and their occurrence together with energies of binding were obtained from the last 60 ns of each trajectory. Besides two types of CSs (IV and VI) authors also investigated role of different ions on the interaction between the protein and CS. Results of their study show that CS-4 forms more stable complexes (ligand is bound stronger to the protein) and Ca2+ ions induce formation of the ionic bridges.
The main drawback of this study is the length of simulations and its convergence. Authors performed only 100 ns long simulations and analyzed last 60 ns. Since the HSA has around 550 residues, the simulations should be considerably longer (~500 ns). Additionally, in order to have statistically meaningful results longer part of the trajectory should be taken into account and not only 60 ns. Finally, stability of the system should also be verified by other means than just based on the energy of the whole system (such as RMSD and RMSF which are not reported at all in the manuscript). Therefore, I would suggest to authors to prolong the simulations considerably and then perform the same analysis as they already did.
Other minor comments:
1) Table 1: Complexes are ranked according to the binding energy. Authors should report numerical values for ranking in this table.
2) Figure 3: I failed to understand how are complexes in this figure ranked from lowest to highest energy of binding as complex 5 with Ca2+ in panel a) has lower energy compared to complex 1 in the same panel.
3) 3D structures of complexes between HSA and CS should be reported and shown in a figure, at least the ones with the most favourable interactions, while the rest should be shown in supplementary information. Ideally, also 2D schemes should be reported.
4) Figure 5 is unclear, there are too many residues shown in balls and sticks representation.
5) HSA should be more clearly described in the introduction section, preferably in a separate figure, where different binding sites are shown (heme binding site, Sudlow's site II etc.)
6) Figure 1: CS4 and CS6 should be written as CS-4 and CS-6 to be consistent with the rest of the text.
7) Line 306: What is "citevanOs"?
8) I believe there is double spacing in Lines 19, 20, 55, 82, 260, 332, and 336.
Reviewer 3 Report
The authors studied the interactions between two synovial fluid (SF) components, human serum albumin (HSA) and chondroitin sulfate (CS). These components are critical to the SF's lubrication properties. The authors applied molecular docking and molecular dynamics methods to study the stability of the HSA:CS complex for two types of CS molecules and, how the addition of different ions (Ca2+, Na+, and Mg2+) affects the different types of interactions occurring in the system.
The presented results are interesting, however, the authors should work on improving the text. The main issue, in my opinion, is how the manuscript was written.
1. Extensive editing of English language and style required: The manuscript needs to be written in a more academic style. There are many difficult-to-understand phrases, for example (page 3), "Moreover, in [31] it was shown, that, during the HA or CS alone significantly lowers the friction torque and dissipated energy of fretting interface and reduce the damage of the articular cartilage, a mixture of HA and CS provided better protection for the cartilage layer".
2. In the materials and methods section, the size of the box does not seem to be correct.
3. It is critical to demonstrate that the studied complexes reached stability during the MD simulation. The authors could include a plot of the RMSD vs time of their simulations, which could possibly be combined with Figure 2.
4. Figure 5 is extremely difficult to grasp. To make it easier to understand what they're trying to show, the atom representations (and some colors) could be changed.
5. Fig 6 to 8 should be put together. The same is true for figures 9 and 10.
6. The Conclusions section should be improved.
Reviewer 4 Report
1. What is the rationale behind choosing 24 subunits to model CS-4 and CS-6 polymers? How did the authors optimized these polymer structures? Authors need to explain the polymer modeling and optimization clearly. This would help the scientific community to follow and reproduce the data in future.
2. There are several missing residues/atoms in the PDB structure of HSA. How did the authors relocate these residues/atoms? Did they use YASARA to model these missing elements. Please provide a brief description on this aspect.
3. Showing first and last snapshots of HSA:CS-6 complexes does not provide perspective on dynamics of HSA. Instead, the authors should initially evaluate the RMSD and RMSF of HSA and compare with HSA:CS-4 system. Further, the authors can compare differential interactions of CS6 and CS4 w.r.t simulation time. Further, the authors can evaluate dynamic structural changes and individual domain motions in HSA as a consequence of CS4/CS6 binding. This information can be included in the suppli. information.
4. The intermolecular HBond analysis suggest that HSA and CS-6/CS-4 have maximal no.of interactions in the presence of Na ions (Fig 6). Can the authors explain this rational behavior in the context of biological relevance?
5. Since the introduction of AMBER ff12SB, new atom types were introduced in AMBER FF. So, I hope the authors aware of this fact and possibly they tested the compatibility between GLYCAM and AMBER14 forcefields.
6. Did the authors choose flexible receptor and ligand approach while docking using AutoDock? Also, did the authors perform biased or unbiased docking?
Round 2
Reviewer 2 Report
Authors addressed majority of the issues raised adequately. However, after reading the manuscript again, one methodological detail remained unclear to me. In Table 1, energies of HSA-CS4/6 complexes are reported but it is not clear on which structure were the energies calculated, on the last snapshot or on the representative structure of the highest populated cluster or averaged over the whole simulations? Ideally, energies should be calculated and averaged over the equillibrated part of the trajectory and reported with standard deviation.
Reviewer 3 Report
There are still numerous errors in the writing (the lack of commas, misspelled words, etc. )
I strongly advise the authors to utilize some text-checking tools.
Reviewer 4 Report
The authors have responded to all my queries, except the following concerns.
1) I believe sum of RMSF plot (Fig. 4) seems unnecessary in the context. What crucial information does it provide? Instead, the authors try to construct a sausage diagram based on RMSF values and try to correlate functionally important regions and their fluctuations w.r.t ligand contacts.
2) Need to improve the grammar throughout the manuscript.
